# A Multi-Implicit Neural Representation for Fonts

**Pradyumna Reddy**[1]                    **Zhifei Zhang**[2]

**Zhaowen Wang**[2]        **Matthew Fisher**[2]        **Hailin Jin**[2]        **Niloy J. Mitra**[1,2]

[1]**University College London**                    [2]**Adobe Research**

## Abstract

Fonts are ubiquitous across documents and come in a variety of styles. They are either represented in a native vector format or rasterized to produce fixed resolution images. In the first case, the non-standard representation prevents benefiting from latest network architectures for neural representations; while, in the latter case, the rasterized representation, when encoded via networks, results in loss of data fidelity, as font-specific discontinuities like edges and corners are difficult to represent using neural networks. Based on the observation that complex fonts can be represented by a superposition of a set of simpler occupancy functions, we introduce *multi-implicits* to represent fonts as a permutation-invariant set of learned implicit functions, without losing features (e.g., edges and corners). However, while multi-implicits locally preserve font features, obtaining supervision in the form of ground truth multi-channel signals is a problem in itself. Instead, we propose how to train such a representation with only local supervision, while the proposed neural architecture directly finds globally consistent multi-implicits for font families. We extensively evaluate the proposed representation for various tasks including reconstruction, interpolation, and synthesis to demonstrate clear advantages with existing alternatives. Additionally, the representation naturally enables glyph completion, wherein a single characteristic font is used to synthesize a whole font family in the target style.

## 1   Introduction

Fonts constitute the vast majority of documents. They come in a variety of styles spanning a range of topologies representing a mixture of smooth curves and sharp features. Although fonts vary significantly across families, they remain stylistic coherent across the different alphabets/symbols inside any chosen font family.

Fonts are most commonly stored in a vector form (e.g., a collection of spline curves) that is compact, efficient, and can be resampled at arbitrary resolutions without loss of features. This specialized representation, however, prevents the adaptation of many deep learning setups optimized for regular structures (e.g., image grids). In order to avoid this problem, custom deep learning architectures have been developed for directly producing vector output but they typically require access to ground truth vector data for training. This is problematic: first, collecting sufficient volume of vector data for training is non trivial; and second, vector representations are not canonical (i.e., same fonts can be represented by different sequence of vectors), which in turn requires hand-coded network parameters to account for varying number of vector instructions.

An alternate approach is to rasterize vectorized fonts and simply treat them as images. While this readily allows using image-based deep learning methods, the approach inherits problems of discretized

35th Conference on Neural Information Processing Systems (NeurIPS 2021).

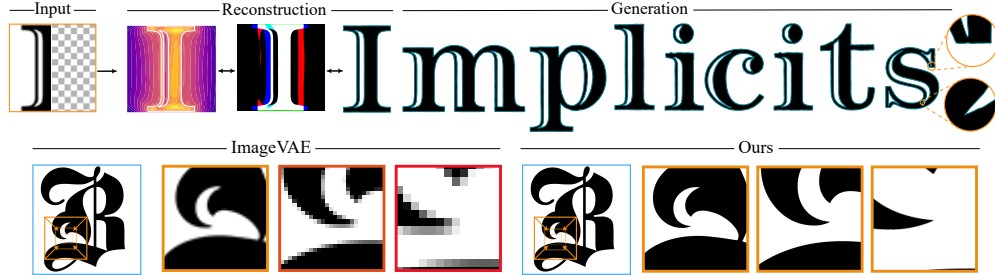

Figure 1: Multi-implicit neural representation for high fidelity font reconstruction and generation. Note that while ours perform similar to ImageVAE at lower/training resolution, the advantage of ours become clear when we test at higher resolution (e.g, how corners continue to be preserved).

representations leading to aliasing artifacts and loss of sharp features. Further, the resultant images are optimized for particular resolutions and cannot be resampled without introducing additional artifacts.

Recently, implicit representation has emerged as an attractive representation for deep learning as, once trained, they can resampled at different resolutions without introducing artifacts. Unfortunately, implicit representations (e.g., signed distance fields) for fonts are often too complex to be represented accurately by neural networks. As a results, although deep implicits work for simple fonts, they can fail to retain characteristic features (i.e., edges and corners) for complex fonts.

Drawing inspiration from multi-channel SDFs [4], we observe that complex fonts can be expressed as a composition of multiple simple regions. For example, a local corner can be represented as a suitable composition of two half-planes, each of which can easily be individually encoded as deep implicit functions. We build on this idea by hypothesizing that complex fonts can also be encoded as suitable composition of global implicit functions. We call such a representation to be a *multi-implicit* neural representation, as each (global) implicit function is neurally encoded.

Thus, multi-implicits provide a simple representation that is amenable for processing by neural networks and the output fidelity remains comparable, even under resampling, to vector representations without losing edge or corner features. A remaining challenge is how to supervise such a network as there is no dataset with reference multi-implicits that be directly used. In this paper, we present a network structure and training procedure that allow multi-implicits to be trained using only *local* supervision. We describe how to extract necessary local supervision from vector input and to adaptively obtain training information for the multi-implicits.

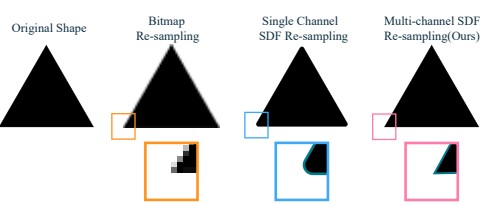

Figure 2: Corner preserving capability of different sampling methods.

We extensively evaluated the proposed multi-implicits representation for various tasks including reconstruction, interpolation, and synthesis to demonstrate clear advantages over several existing state-of-the-art alternatives. Additionally, the representation naturally enables glyph completion, wherein a single characteristic font glyph is used to synthesize a whole font family in consistent style.

## 2 Related Work

**Raster-based representation.** One of the most intuitive representation of shapes is raster, i.e., representing a 2D shape by pixels or a 3D shape by voxels, which provides the grid-format data that is perfectly compatible with (regular) CNN models. Hence it has acted as a catalyst for many deep learning based methods for semantic editing of raster-based shapes. A general idea is to learn an encoding-decoding model and then manipulate the shape in the latent space. Sharing a similar spirit, many generative models [21, 5, 15] engaged in the raster-based generation and attribute transfer. [1, 25] narrowed down the scope to fonts specifically, which focus more on the shape instead of texture. While the raster-based representation has shown strong semantic editability as incorporated

with deep models, it is still limited by its intrinsic resolution. Since image super-resolution is an ill-posed problem, details cannot be fully recovered after upscaling.

**Deep learning based vector graphs.** Vector graphics (e.g., SVG) have been widely adopted as a scalable representation. Typically, it is constructed by sequences of Bézier curves, which are difficult to be modeled by traditional CNN models since the diverse sequence length across different shapes, as well as diverse types and numbers of Bézier curves for the similar or even the same shape. Therefore, RNN-based models (e.g., LSTM) are commonly adopted in recent works [17, 2] to learn the dynamic sequence of curves. Although the sequential modeling could achieve semantic editing and/or interpolation between shapes, it is still lagging behind CNN-based models (raster-based methods) in terms of reconstruction accuracy because it has to model much longer temporal dependency. Also, there is no specific attention given to features like edges and corners.

**Transformation from raster to vector.** Instead of directly modeling raster or vector, an idea of achieving both scalability and editability is to model the transfer from raster to vector [20] (we will not consider vectorization methods that purely transfer images to vectors). Such an approach inherits the advantages of raster-based methods and supports easy editability. In addition, they output scalable vectors directly. However, this method would be still limited by the raster resolution, i.e., the output vector graph cannot capture enough details since the corresponding raster input may have already lost those finer details. Another challenge for such methods is the high complexity of the shape, e.g., interpolation between shapes with different topologies leads to undesirable intermediate shapes.

**Deep implicit representation.** Deep implicit functions [10, 3, 8, 22, 18, 23] have achieved great success in shape representation. They take advantage of deep learning techniques to fit an implicit function, which provides a continuous representation breaking the grid limitation of a raster domain. In addition, deep implicit functions model shapes spatially instead of modeling sequentially as afore-mentioned in deep learning based vector graphs. Therefore, deep implicit functions could preserve the editability like raster-based representation and potentially achieves scalable representation like vectors. Unfortunately, existing works seldom explore the scalability and fidelity of shapes in their representations, especially during editing, interpolation, and upscaling. For instance, sharp corners always suffer from editing and scaling. This is particularly problematic in the domain of fonts, and our work addresses this limitation via the proposed multi-implicit representations.

## 3   Methods

We represent fonts as the composition of multiple global implicit functions. An implicit representation has two key advantages: it allows for rendering at arbitrary resolution; it can be locally supervised in characteristic areas such as sharp edges and corners. Unlike a single implicit function, our multi-implicit representation can faithfully reconstruct sharp features with low reconstruction error (Figure 2).

We train a generative neural network, as shown in the inset, that models fonts using this representation. Instead of directly learning the inside-outside status of the font image [3], we predict a set of distance fields of 2D shapes. Their com-

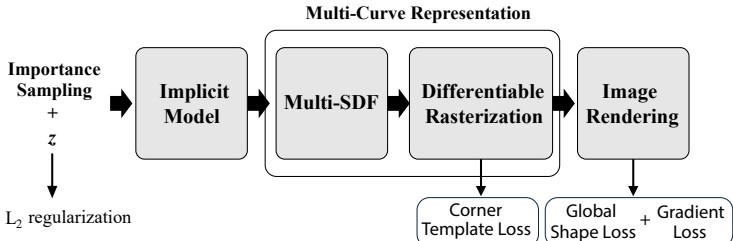

position is then fed into a differentiable rasterizer to produce the final image. We will show in Section 4 how this generative model enables a diverse set of font reconstruction and editing operations. Section 3.1 details differentiable rasterization of 2D distance fields, and corresponding corner preservation is discussed in Section 3.2. Finally, training losses and details are given in Section 3.3.

### 3.1   Differentiable Rasterization of Distance Field

As aforementioned, we use the signed distance field (SDF) to model 2D shapes. The common supervisions for SDF are SDF labels from the ground truth curves or distance transform on the silhouette [14]. Figure 3 compares the results between training with SDF and training with raster,

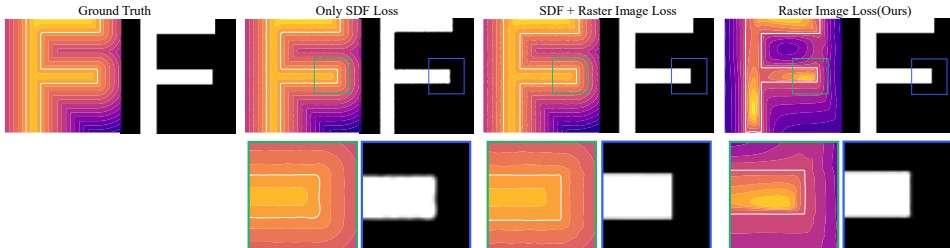

| Ground Truth | Only SDF Loss | SDF + Raster Image Loss | Raster Image Loss(Ours) |

Figure 3: **Reconstruction with different training signals.** Training using SDF (2nd column) does not ensure good reconstruction in raster domain, while training with both SDF and rasterized image (3rd column) and only rasterized SDF (ours) yields smoother boundary.

where training only on the ground truth SDF does not always result in a good raster image in terms of boundary smoothness. In contrast, training with the supervision of raster yields smoother boundary. Therefore, we will draw supervision on rasterized SDF to achieve better shape.

Sharing the spirit from vector graphics rasterization works [7, 11, 16, 19, 24], we simplify general vector graphics rendering by analytically approximating the point to curve distance as,

$$I(x, y) = K\left(\min_{i \in \mathcal{F}} d_i(x, y)\right) g(x, y),$$

(1)

$$K(d) = \begin{cases} 1 & \text{if } d > \gamma, \\ k\left(\frac{d}{\gamma}\right) = \frac{1}{2} + \frac{1}{4}\left(\left(\frac{d}{\gamma}\right)^3 - 3\left(\frac{d}{\gamma}\right)\right) & \text{if } -\gamma \le d \le \gamma, \\ 0 & \text{if } d < -\gamma, \end{cases}$$

(2)

where $d_i(x, y)$ indicates an SDF of distance from pixel $(x, y)$ center to the closest point on the $i$-th curve corresponding the font $\mathcal{F}$, and $K$ denotes a function that approximates the opacity based on the distance value. There are many works for estimating $d_i(x, y)$ from scene parameters but most of them are constrained by the choice of the parameterization. In this paper, we model each $d_i(x, y)$ using an implicit neural network. The function $g$ models the spatially-varying texture of the shape. For solid fonts, we set $g = 1$. In the supplemental we show examples of textured fonts where we make use of spatially varying $g$. In the function $K$, $\gamma$ is the anti-alias range, and the kernel $k$ is a radially symmetric continuous filter that satisfies the constraint $k(1) = 0$ and $k(-1) = 1$. In our work, we approximate $k(\cdot)$ using a parabolic kernel similar to [19, 13]. Note that the rasterization function $I(x, y)$ has non-zero gradients only if $(x, y)$ falls inside of the anti-aliasing range. We use a progressively decreasing anti-aliasing range strategy for better convergence and fidelity (see Sec. 3.3).

## 3.2   Multi-Curve Representation for Sharp Corners

We will lose details like sharp corners when upscaling bitmaps or sign distance functions. Resampling an implicit model that encodes the pixel values or signed distance values of a shape similarly suffers from blurry corners. An brute force solution is to train the implicit model with extremely high-resolution images, but this would drastically increase the burden of training, and still limited by the training resolution. Rather than

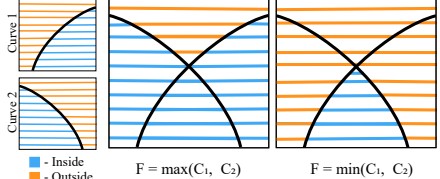

(a) Possible corner preserving shapes represented with n=2.

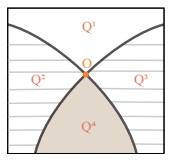

(b) Corner template.

Figure 4: Intersection of two curves to encode concave and convex corners.

directly modeling corners, we represent corners as the intersection of multiple curves (e.g., lines or parabolas), drawing inspiration from traditional representations [9]. Note that even though these individual sub-curves may be smoothed after encoded by a deep model, the sharpness of their intersection will be preserved. With this insight, we construct sharp corners from multiple smooth curves

predicted by the implicit model. More specifically, the implicit model is designed to predict multiple SDFs (and rasterization of distance fields), each of which carries smooth curves/shapes decoupled from corners and edges as illustrated in Figure 4a.

Assume a shape $\mathcal{F}$ is represented by a set of curves $\mathcal{C} = \{C_1, C_2, \cdots, C_n\}$, where $C_n$ is a binary map indicating whether a pixel is inside (i.e., 1) or outside (i.e., 0) the $n$-th curve, like the example in Figure 4a. In our scenario, $C_n = K(d_n)$, where $d_n$ is the $n$-th SDF channel estimated by the implicit model. Then, a function over all curves $F(\mathcal{C})$ will fuse those curves to reconstruct the shape, preserving sharp corners. As illustrated in Figure 4a, two curves can sufficiently represent a corner, either convex or concave by adopting maximum or minimum as the function $F$. To represent a shape with arbitrary corners, however, it requires three curves at least. For example, $F(\mathcal{C}) = \min(\max(C_1, C_2), C_3)$ can model all corners in a shape. Therefore, we set $n = 3$, i.e., $\mathcal{C} = \{C_1, C_2, C_3\}$. Since deep models are sensitive to the permutation of guidance signals during training, we use the median function as $F$, which achieves sharp corners and permutational invariance to the order of these curves. Thus, a corner-related loss on $F(\mathcal{C})$ could be robust to the ordering ambiguity.

Based on $F(\mathcal{C})$ (median function on three curves), a typical corner $O$ is presented in Figure 4b, where intersection of two curves divides its local space into four quadrants, i.e., from $Q_1$ to $Q_4$. There are always two opposite quadrants that one is inside area (i.e., $Q_1$) where corresponding values from $F(\mathcal{C})$ are 1, and the other is outside area (i.e., $Q4$) where corresponding values from $F(\mathcal{C})$ are 0. The rest two opposite quadrants (i.e., $Q_2$ and $Q_3$) are equal on $F(\mathcal{C})$ but different on $\mathcal{C}$. For example, the values on $\{C_1, C_2, C_3\}$ corresponding to the $Q_2$ area is $(1, 0, 0)$, so $F(\mathcal{C})$ on $Q_2$ is 0. Then, $F(\mathcal{C})$ on $Q_3$ should be 0 as well. However, the values on $\{C_1, C_2, C_3\}$ corresponding to the $Q_3$ region must be different from $(1, 0, 0)$, i.e., could be $(0, 1, 0)$ or $(0, 0, 1)$. If $F(\mathcal{C})$ on $Q_2$ and $Q_3$ is 1, $O$ is a concave corner. Otherwise, $O$ is a convex corner. Such distribution of $\mathcal{C}$ along the four quadrants around a corner $O$ is referred to as corner template.

The shapes are encoded as the multi-curve representation through an implicit model. The implicit model takes sample $(x, y, l)$ (i.e., 2D spatial location $(x, y)$ and glyph label $l$), as well as embedding $z$ that indicates the font style, and outputs three channels of SDF $\{d_1, d_2, d_3\}$. Then, the rasterization approach discussed in Differentiable Rasterization of Distance Field converts each $d_i$ to $C_i$ ($i = 1, 2, 3$). Finally, the median function $F(\mathcal{C})$ renders the final shape. We optimize the global shape using the final render output and (locally) supervise each corner to be locally represented as an intersection of two curves. One could use the output of [4] to train a network. However, the edge coloring approach presented in [4] has no canonical form, which prevents a neural network from learning a continuous latent space between shapes. Please note that since we focus on corner supervision to ensure sharp corners at higher resolution resampling, it is unnecessary to constraint the model by multi-channel supervision globally.

In the optimization of rendered global shape, the median operation $F(\mathcal{C})$ would route the gradients to the correspondingly active value only, i.e., only update a single channel at each location of $\mathcal{C}$. However, at least two of the three channels of $\mathcal{C}$ at a certain location need to be updated to approaching the ground truth because of the nature of the median operator. Therefore, in the training stage, we use an approximation $\hat{F}(\mathcal{C})$ defined as the average of the median and the closest value to the median, thus two channels will be updated.

### 3.3 Training Details

We use three losses on the shape of glyph: (i) a global shape loss that captures glyph shapes globally, (ii) corner template loss that supervises intersection of curves locally to make the shape robust against resampling and editing; and (iii) Eikonal loss to adhere to true SDFs.

For global shape training, since gradients are non-zero only at the anti-aliasing range, i.e., edges of the shape, we sample the edges of the rasterized glyph to train the implicit model. We shape $3 \times 3$ neighborhoods around the anti-alias pixel, where we have a sample of one value outside the shape (i.e., 0), one value inside the shape (i.e., 1), and everything in between. Meanwhile, we sample from homogeneous areas inside and outside of the glyph, such that the model does not fit a degenerate solution. The edge/corner-aware sampling is referred to as

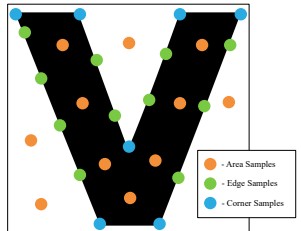

importance sampling (see inset). Such non-standard sampling further motivates the use of implicit models, and importance sampling would significantly reduce the computational complexity for training on higher resolution shapes as compared to the traditional training on grid images.

We measure global shape loss between the final rendering from $F(\mathcal{C})$ and rasterized glyph $I$. We use a differential approximation $\hat{F}(\mathcal{C})$ at training time. All the edges, area, and corner samples from the raster image are used to train the implicit model via mean square error as,

$$\mathcal{L}_{global} = \mathbb{E}\left(\hat{F}(\mathcal{C}) - I\right)^2 . \tag{3}$$

For local corner template loss, we first perform corner detection. A corner is defined as a local where two curves intersect at an angle less than a threshold (the threshold is 3rad or $171°$ in our experiments). For each corner, we generate the corner template as discussed in section 3.2. The template size is $7 \times 7$ corresponding to the image size of $128 \times 128$. The size of the corner template is scaled based on the size of the image, but the size of the sampling neighborhood for the global shape training remains the same. We densely sample the edges and corners, and sparsely sample the homogeneous areas.

To represent glyph corners as the intersection of two curves, we supervise the corner samples by the corresponding corner templates. Since the render function $F(\mathcal{C})$ is invariant to the order of the SDF/raster channels, the corner template loss inherits the permutation invariance to the channel order and, in order to avoid unnecessarily constraining the network, we only supervise $Q_2$ and $Q_3$ of the template using the loss,

$$\mathcal{L}_{local} = \mathbb{E}_{O \in \text{corner samples}} \sum_{i=2}^{n} \min_{j \in \{2,\cdots,n\}} \left(C_i^O - T_j(O)\right)^2 , \tag{4}$$

where $O$ indicates a corner sample from ground truth, and $n$ is the number of channels (i.e., $n = 3$ in our setting). The correspondingly predicted curves of the corner $O$ are denoted by $C_i^O$, and the corresponding corner template is $T(O)$ that has $n$ channels indexed by $j$.

The gradient loss aims to constraint the output of the implicit network to resemble a distance field this is so that the implicit re-sampling is more well behaved and resembles a closed continuous shape. A special case of Eikonal partial differential equations [6] posits that the solution to $d(x, y; \theta)$ must satisfy $\mathbb{E}||\nabla d(x, y; \theta)|| = 1$, where $d(x, y; \theta)$ denotes a simplified implicit model parameterized by $\theta$. Since satisfying this constraint is not completely necessary for our desired solution, we loosen it to be greater than or equal to 1 as Eq. 5, which intuitively encourages the function to be monotonic.

$$\mathcal{L}_{grad} = \begin{cases} \mathbb{E}|1 - ||\nabla d(x, y; \theta)||_2| & \text{if } ||\nabla d(x, y; \theta)||_2 < 1, \\ 0 & \text{if } ||\nabla d(x, y; \theta)||_2 \geq 1, \end{cases} \tag{5}$$

where $||\cdot||_2$ represents the $\ell_2$-norm, and $|\cdot|$ calculates the absolute values. Finally, the total loss is

$$\mathcal{L} = \mathcal{L}_{global} + \alpha\mathcal{L}_{local} + \beta\mathcal{L}_{grad} + \gamma||z||_2, \tag{6}$$

where $\alpha$, $\beta$, and $\gamma$ are weights to balance these terms during the training.

**Training Warm-up.** Since the gradients mainly fall into the anti-aliasing range, network initialization would significantly affect the convergence. To this perspective, we set the initial anti-aliasing range to be the whole image range and slowly shrink it to $k \cdot w^{-1}$ during the training, where $w$ is image width, and $k = 4$ in our experiments. Such warm-up helps the model converge more consistently, and the estimated SDF is more well behaved. Comparison of SDF with and without warm-up is conducted in the supplementary.

## 4  Experiments

We evaluate our method against the tasks of reconstruction, interpolation, and generation. In reconstruction and interpolation, we compare our method to ImageVAE [12], DeepSVG [2], and Im2Vec [20], while we compare to DeepSVG and Attr2Font [25] in the generation task. The metrics for evaluating the rendered glyphs are mean squared error (MSE) and soft IoU (s-IoU), which is defined as

$$\text{s-IoU}(I_1, I_2) = ||I_1 I_2||_1 / ||(I_1 + I_2)_{|0,1|}||_1, \tag{7}$$

Table 1: Comparison with baselines on reconstructing training samples at different resolutions. In training, we use resolution of $64 \times 64$. In testing, to achieve target resolution, we bilinearly upsample ImageVAE output from 64, rasterize vector from DeepSVG and Im2Vec, and directly query ours.

| | MSE ↓ | | | | s-IoU ↑ | | | |
|---|---|---|---|---|---|---|---|---|
| Methods | 128 | 256 | 512 | 1024 | 128 | 256 | 512 | 1024 |
| ImageVAE | .0072 | .0120 | .0160 | .0186 | .8252 | .8416 | .8482 | .8494 |
| DeepSVG | .1022 | .1081 | .1108 | .1121 | .3073 | .3124 | .3162 | .3164 |
| Im2Vec | .0435 | .0518 | .0557 | .0571 | .7279 | .7293 | .7294 | .7294 |
| Ours | .0118 | .0170 | .0201 | .0218 | .8750 | .8978 | .9035 | .9049 |

where $I_1$ and $I_2$ are the images to compare, $\| \cdot \|_1$ denotes the $\ell_1$-norm, and $\lfloor 0, 1 \rceil$ clips the values to the interval of [0,1]. To evaluate the fidelity of glyph rendering in larger scales, glyphs are rendered at the resolution of 128, 256, 512, and 1024 from each method without changing the training resolution ($64 \times 64$).

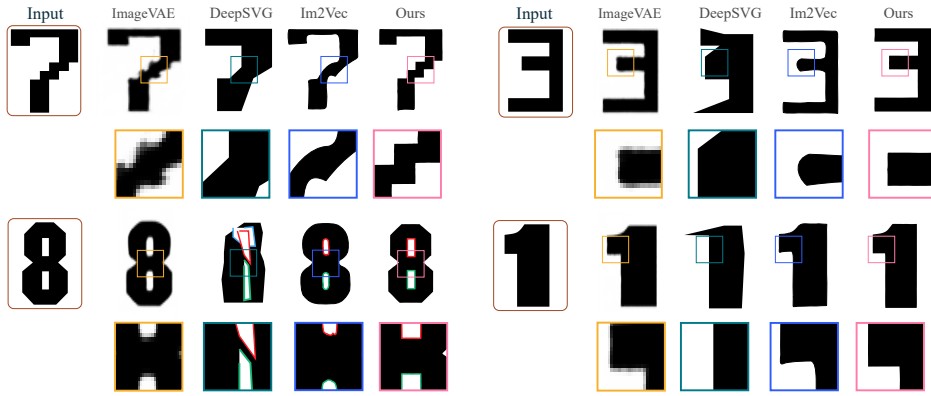

Figure 5: Reconstruction examples (baseline vs. ours) with zoom-in box highlighting corners. We have vectorized the zero-level-set of the SDF output in a piece-wise linear way. Please use digital-zoom to take a closer look at the difference in reconstruction quality.

**Reconstruction and Interpolation.** We compute MSE and s-IoU over the training dataset at different resolutions to quantify how different algorithms capture the input training dataset. For a fair comparison, we train all the algorithms on the same dataset used by Im2Vec [20], which consists of 12,505 images. Table 1 displays MSE and s-IoU metrics on the training set. For ImageVAE, we perform bilinear interpolation to obtain higher resolution outputs. Our method outperforms the others on s-IoU, indicating better reconstruction of the glyph shapes. ImageVAE gets higher scores on MSE because its training objective aligns with the MSE metric. However, ImageVAE would show blurry shapes in interpolation and editing as demonstrated in Figure 6a, where we achieve more continuous interpolation/latent space. Even in reconstruction, as visualized in Figure. 5, ImageVAE cannot preserve sharp boundary and corners as compared to the other methods. Since DeepSVG and Im2Vec directly output vectors, they can always render shapes with clear boundaries, but they are limited in capturing the global shapes as compared to our method. Another advantage of our method over DeepSVG and Im2Vec is that we learn a smoother latent space, achieving better performance on interpolation as demonstrated in Table 2. Even interpolating between complex shapes, as shown in Figure 6b, our method performs better than the state-of-the-art Im2Vec. In Table 2 we present the MSE and s-IOU calculated between a random interpolated glyph and its nearest neighbour in the training dataset. This helps quantify similarity between training versus generation distribution.

We have also experimented with Fourier features, Sine activation and Sawtooth activation function with a feed forward network, due to the high frequency nature of these functions the latent space learned by such a network is not continuous. Since learning a continuous latent space is essential for applications like interpolation and font family generation from a complete or partial glyph, we resorted to using a LeakyReLU based activation function. However if the users main requirement is only faithful reconstruction of the input dataset, single channel fitting with fourier features trained

using our differential rasterization function should yield similar results as multi channel fitting with LeakyReLU activation.

Table 2: Comparison with baselines on interpolation at different resolutions.

| Methods | MSE ↓ | | | | s-IoU ↑ | | | |
|---|---|---|---|---|---|---|---|---|
| | 128 | 256 | 512 | 1024 | 128 | 256 | 512 | 1024 |
| ImageVAE | .0181 | .0183 | .0185 | .0185 | .7715 | .7721 | .7731 | .7734 |
| DeepSVG | .0544 | .0556 | .0569 | .0575 | .6337 | .6347 | .6365 | .6372 |
| Im2Vec | .0434 | .0445 | .0463 | .0473 | .7213 | .7218 | .7232 | .7238 |
| Ours | .0279 | .0297 | .0316 | .0343 | .8181 | .8184 | .8222 | .8234 |

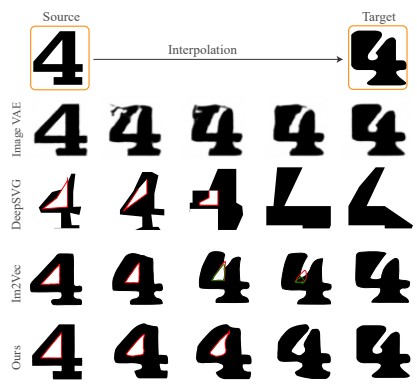

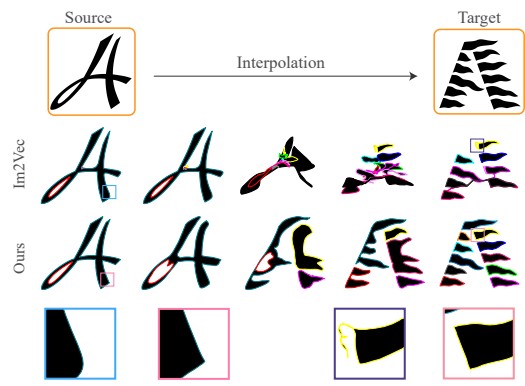

(a) Comparison of all methods.  (b) A challenging example where baselines fail.

Figure 6: Comparison of interpolation between two random font styles. We color different curves in the vector output to highlight details.

**Generation.** To generate new fonts and the corresponding glyphs, first the auto-decoder implicit model is trained with latent vector $z$ and glyph label (i.e., one-hot encoding) concatenated to spatial locations as the input. We train on 1,000 font families, i.e., 52,000 images, and test on 100 font families. In the inference stage, given an unseen glyph, we first find the optimal latent vector (i.e., font style) that makes the rendered glyph closest to the given glyph. More specifically, fixing the glyph label based on the given glyph, its font latent vector $\hat{z}$ can be obtained by minimizing the distance between the raster of the given glyph and the predicted glyph using gradient descent. With the optimal $\hat{z}$, all the other glyphs with the same font style can be generated by iterating the glyph label. Figure 7 compares the font completion results between the baselines and ours, where a glyph "A" is given with unseen font style. Our results outperform the others in terms of global shape and sharpness of boundaries and corners. In general, raster-based methods (e.g., Attr2Font) tend to generate better shape but get blurry at corners. By contrast, vector-based methods (e.g., DeepSVG) eliminate the blurry effect while difficult to achieve good global shapes. It is a dilemma of generating better global shapes or better local corners in recent works. Our method is achieving both good shapes and corners. Table 3 provides statistical results that further demonstrates the superior generation capacity of our method.

We conduct a more challenging task, i.e., glyph completion, to explore the potential generation capacity of our method. As illustrated in Figure 8, given a partial glyph, it can still recover the whole glyph, as well as other glyphs with the same font style. In addition, sharp corners are still preserved.

Table 3: Comparison with baselines on font generation task at different resolutions.

| Methods | MSE ↓ | | | | s-IoU ↑ | | | |
|---|---|---|---|---|---|---|---|---|
| | 128 | 256 | 512 | 1024 | 128 | 256 | 512 | 1024 |
| DeepSVG | .2597 | .2768 | .2854 | .2911 | .3584 | .3613 | .3651 | .3672 |
| Attr2Font | .2004 | .2231 | .2481 | .2563 | .6204 | .6451 | .6523 | .6560 |
| Ours | .0946 | .1027 | .1065 | .1083 | .8429 | .8462 | .8469 | .8471 |

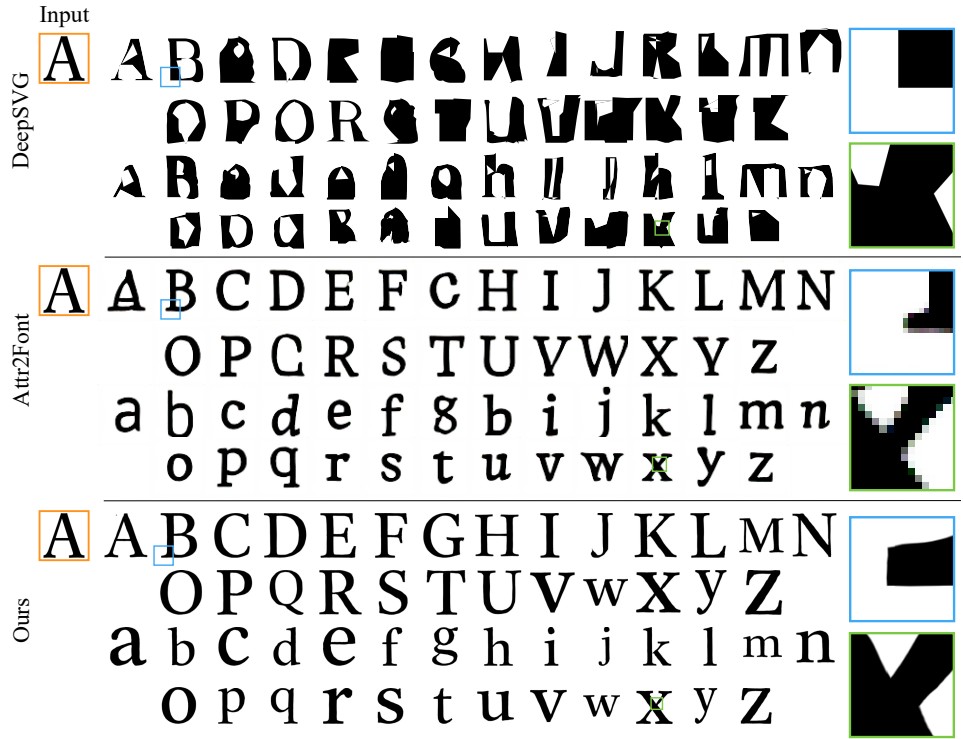

Figure 7: Font completion examples (baseline vs. ours) with zoom-in boxes highlighting the corners.

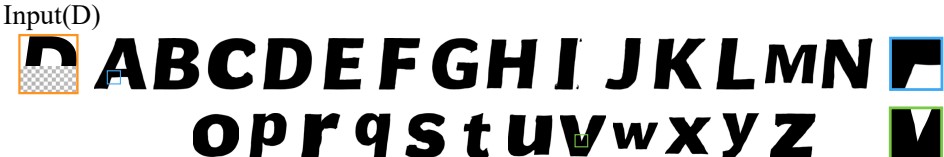

Figure 8: Glyph completion example. Given a partial glyph unseen in the training set, our method can complete the given glyph and other glyphs with the same font style. The zoom-in boxes highlight the corners. The mask region in the input is ignored during optimizing the latent vector $\hat{z}$.

## 5   Conclusion

We have presented multi-implicits – a new vector representation that is easy to process with neural networks and maintains 2D shape fidelity under arbitrary resampling. The representation is learned in a locally supervised manner allowing high precision recovery of corners and curves. The proposed multi-implicits representation is extensively evaluated in various font applications, including high-resolution reconstruction, font style interpolation, font family completion, and glyph completion, and demonstrates clear advantages over prior image based and curve based approaches.

**Broader Impact**   The proposed representation has the potential to be applied to other 2D vector objects, such as icons and animations, which can empower artists' creativity and productivity. If trained on a handwriting dataset such a method could possibly be used for emulating a person's handwriting for forgery.

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
