# A Multi-Implicit Neural Representation for Fonts

# Supplementary

## S.1 Network Details

In our experiments, we train an auto-decoder based network which is an 8-layer MLP, and each hidden layer contains 384 neurons. We use the LeakyReLU activation function as the non-linearity. The latent embedding $z$ is a 128-D vector. For better convergence, sharing the spirit from [4], a skip connection is built between inputs and the third hidden layer, i.e., the inputs are concatenated to the output of the third hidden layer. In the training, the ADAM optimizer [3] is adopted with the learning rate $10^{-3}$, and the network is trained for 2000-3000 epochs. Rather than following the traditional training routine in the reconstruction and interpolation tasks, the training strategy for the generation task is to freeze the learned latent embedding weights after 1000 epochs, such that the training is more stable across glyphs of the same font family.

We also experimented with sine activation function [5] and high-frequency Fourier feature positional encoding [6]. However, both of them do not learn a continuous latent space to perform generative tasks as discussed in the main paper (Section 4). Therefore, we choose LeakyReLU over the alternatives.

## S.2 More Qualitative Results

We also include more results on reconstruction, interpolation, full glyph input generation, and partial glyph input generation. All the images presented are rendered at 4x - 16x of their training resolution. Among the shared results, we have hand-picked interesting font family reconstruction results. However, the interpolation, full glyph input generation, and partial glyph input generation results are randomly sampled. This combination should give the reader a sense of the capabilities of our pipeline. We show samples of the results in Figures S.1, S.2, S.3, S.4, and S.5 (link to download more results).

## S.3 Ablation study.

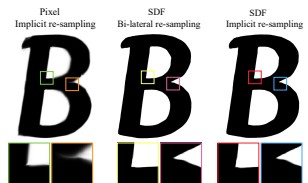

The key to achieving sharp corners in our method is the multi-channel SDF representation, which models multiple curves to better construct corners. An alternative can be to directly supervise multi-channel signals by raster, i.e., removing the intermediate SDF representation. We compare the two methods in Table S.1, where SDF implicit re-sampling and pixel implicit re-sampling denotes the methods with and without SDF representation, respectively.

The inset shows ablation results where the SDF implicit re-sampling achieves sharper corners. For re-sampling higher resolution images, our method provides two ways: 1) implicit re-sampling and 2) bilateral re-sampling. The implicit re-sampling performs dense sampling through the implicit model to obtain higher resolution SDF and thus rendering higher resolution shapes. Bilateral re-sampling will directly upsample the SDF prediction from the implicit model to the target resolution.

Table S.1: Ablation study on reconstruction quality at different resolutions and sampling strategies.

| | MSE ↓ | | | | s-IoU ↑ | | | |
|---|---|---|---|---|---|---|---|---|
| Methods | 128 | 256 | 512 | 1024 | 128 | 256 | 512 | 1024 |
| Pixel implicit re-sampling | .0135 | .0172 | .0192 | .0203 | .8335 | .8521 | .8574 | .8594 |
| SDF bilateral re-sampling | .0118 | .0171 | .0202 | .0219 | .8761 | .8990 | .9033 | .9046 |
| SDF implicit re-sampling | .0118 | .0170 | .0201 | .0218 | .8750 | .8978 | .9035 | .9049 |

## S.4  Code and Data Preparation Details

We include a development version of our code in the supplemental folder to help the reader reproduce our results. The README file in the code folder contains the link to a sample dataset and instructions to train a model. The included local_eval_family python notebook helps the reader to generate results similar to the ones presented in the paper.

In our implementation, we detect corners using the directly available vector graphics information. In practice, however, one can always use methods like [2] where they report 99.3% corner detection accuracy using random forests. For the sake of implementation convenience, we use [1] to generate the MSDF representation and mask the neighborhood around the detected corners to generate corner templates.

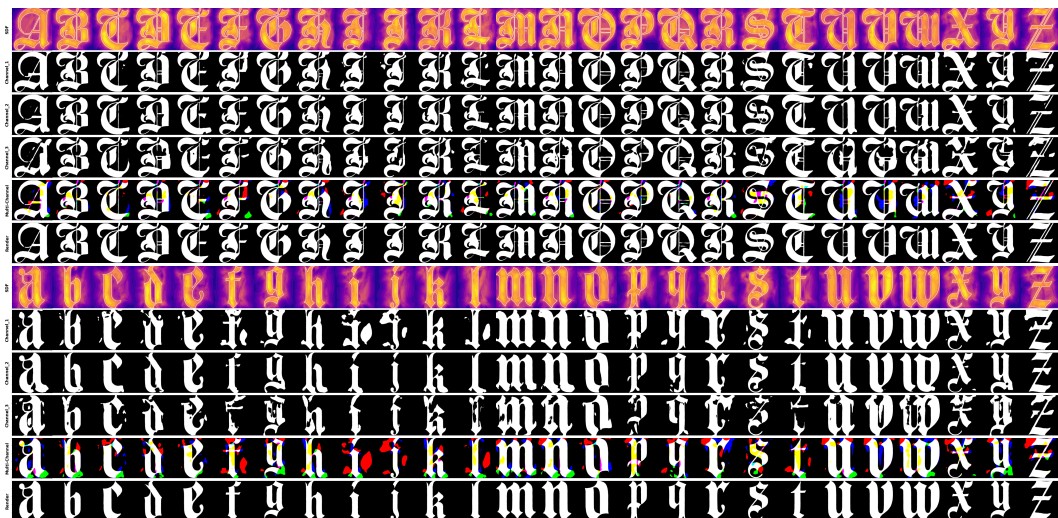

Figure S.1: **Reconstruction** Each reconstruction result we present contains 5 rows. The first row corresponds to the estimated SDF, followed by individual channels in the multi-channel representation. All the channels of the multi-channel representation combined and visualized in the fourth row. Finally, in the last row, we show the reconstruction render.

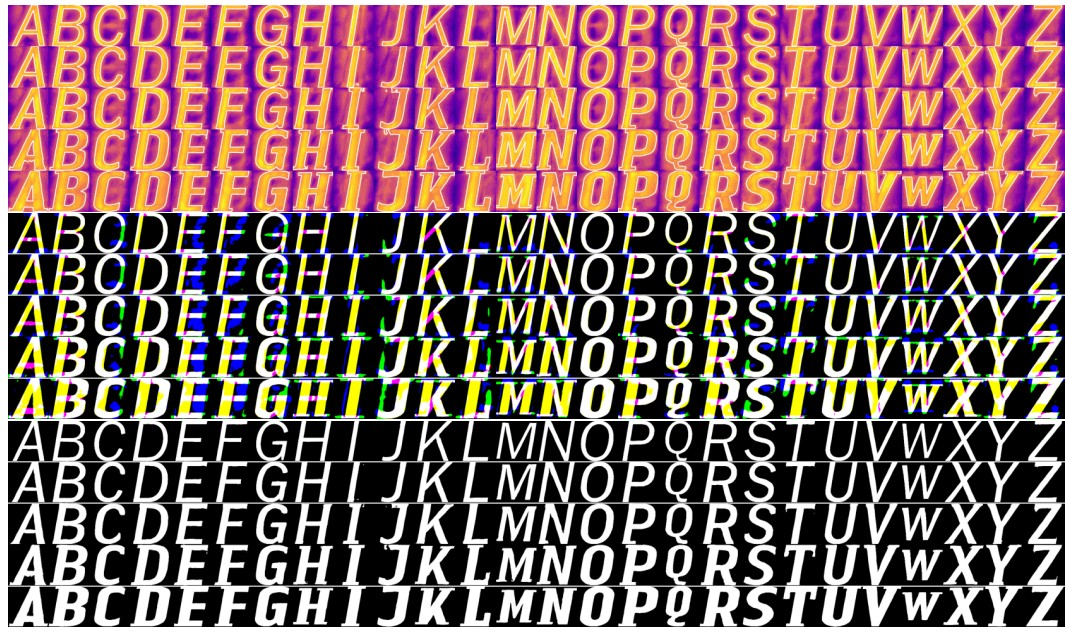

Figure S.2: **Interpolation (upper case)** Here we visualize the capital alphabets of a font family. Each interpolation result we present is divided into 3 files corresponding to SDF, multi-channel representation, and final renders. Each file contains 5 rows showing linear interpolation between font family in the top to font family in the bottom in the learned latent space.

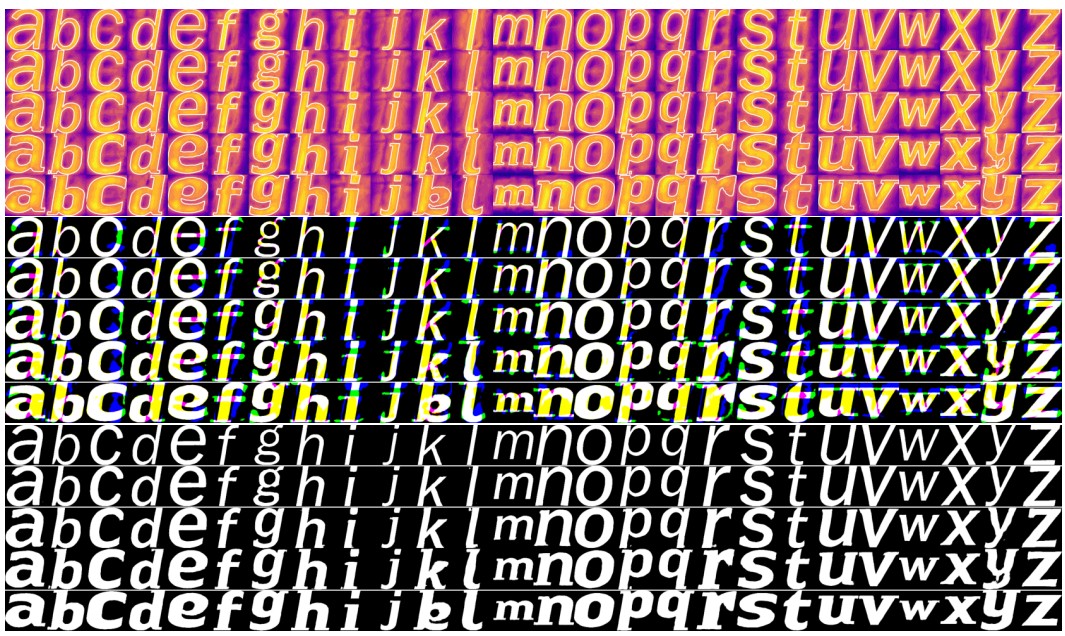

Figure S.3: **Interpolation (lower case)** Similar to the interpolation results of the capital letters of a font family the interpolation of the small letters is also divided into 3 files corresponding to SDF's, multi-channel representation, and final renders. Each file contains 5 rows showing linear interpolation between small alphabets of the font family in the top to small alphabets of the font family in the bottom in the learned latent space.

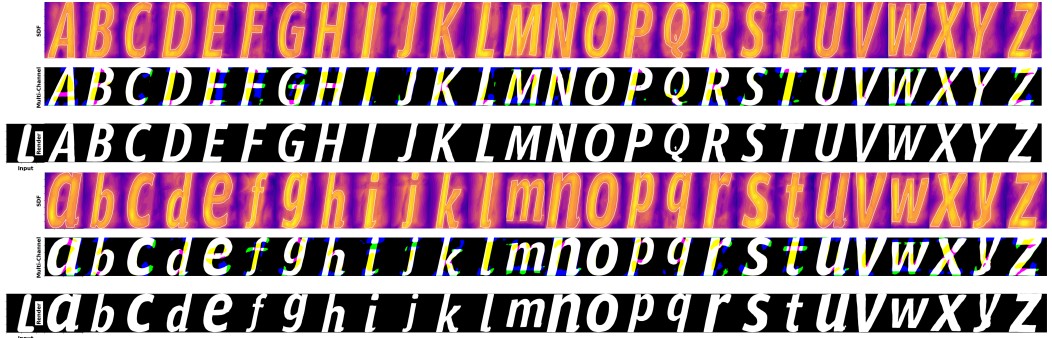

Figure S.4: **Generation (from full glyph)** For each file in the generation_full folder we show the input glyph in the first column and the estimated SDF, multi-channel representation, and final renders in the other columns.

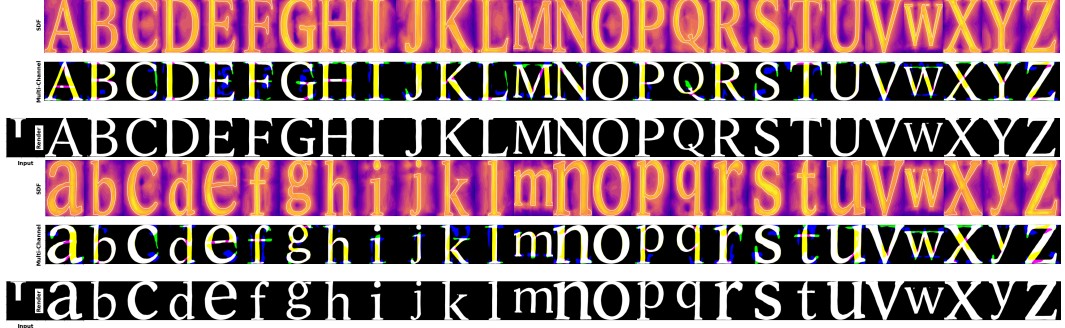

Figure S.5: **Generation (from partial glyph)** Similar to generation_full for each file in the generation_partial folder we show the input partial glyph in the first column followed by the estimated SDF, multi-channel representation, and final renders in the other columns.

## S.5 Corner Reconstruction

We evaluate the reconstruction performance specifically on the corner regions, using the same metrics as shown in Table S.3. Similar to other quantitative results in the main submission, our method outperforms the baselines in terms of s-IoU, which focuses on the foreground (i.e., glyph) and ignores the irrelevant background, thus better measuring corner accuracy. In training, we use the resolution of $64 \times 64$. In testing, to achieve target resolution, we bilinearly upsample ImageVAE output from 64, rasterize vector from DeepSVG and Im2Vec, and directly query from ours. The testing samples are from the training set.

Table S.2: Comparison with baselines on reconstructing corner regions at different resolutions.

| | MSE ↓ | | | | s-IoU ↑ | | | |
|---|---|---|---|---|---|---|---|---|
| Methods | 128 | 256 | 512 | 1024 | 128 | 256 | 512 | 1024 |
| ImageVAE | .0012 | .0021 | .0026 | .0028 | 0.7495 | 0.7497 | 0.7498 | 0.7498 |
| DeepSVG | .0324 | .0344 | .0353 | .0358 | 0.6893 | 0.6895 | 0.6896 | 0.6897 |
| Im2Vec | .0136 | .0163 | .0175 | .0180 | 0.7469 | 0.7470 | 0.7471 | 0.7471 |
| Ours | .0057 | .0083 | .0098 | .0107 | 0.8838 | 0.8846 | 0.8848 | 0.8849 |

## S.6 General Rasterization

We worked with solid (black and while) fonts for the reconstruction, interpolation, and generation tasks in our experiments. Beyond that, our method can be extended to textured fonts by parameterizing the function $g$ in Eq. S.1 (recap Section 3.1 in the main submission),

$$I(x, y) = K\left(\min_{i \in \mathcal{F}} d_i(x, y)\right) g(x, y). \tag{S.1}$$

In this experiment, we parameterize the function $g$ as an implicit neural network with a similar configuration as the distance fields. However, unlike the distance field network for $d$, we use a sine activation function, which could better model the high-frequency feature of texture in the input images. Since the parameterization has many possible factorizations, we add two extra losses to our loss function to encourage a minimal and smooth/continuous alpha mask. More specifically, one loss is $\mathcal{L}_{alpha} = \|\mathcal{C}\|_1$ that encourages a minimal alpha mask but might result in a grainy mask, so the other loss is $\mathcal{L}_{div} = \Delta d$, i.e., minimizing the divergence of the distance field to achieve more smooth/continuous alpha mask. As illustrated in Figure S.6, our method can well represent shapes with internal texture, demonstrating the strong potential of adapting our method to more general graphs.

## S.7 Canonical Multi-Channel Representation

Figure S.7 shows the multi-channel representation estimated by a neural network trained using our pipeline and multi-channel representation generated by [1]. Please note that global supervision with multi-channel representation generated by [1] would not lend itself to smooth interpolation between glyphs and fonts.

## S.8 Limitations

Our method has a couple of limitations because of the nature of the encoding and raster-based training procedure. If a curve length is small and the curve ends in a convex and concave corner, the corresponding corner template would carry higher frequency information that may be too high to be represented correctly by a LeakyReLU based neural network. This can be addressed by training at higher resolution inputs and/or increasing the network capacity at the cost of increased computation time. Secondly, if training at a lower resolution and re-sample at an extremely high resolution, small fitting errors are accentuated leading to wobbly lines.

| Input | SDF | Alpha | Texture | Render |
|-------|-----|-------|---------|--------|

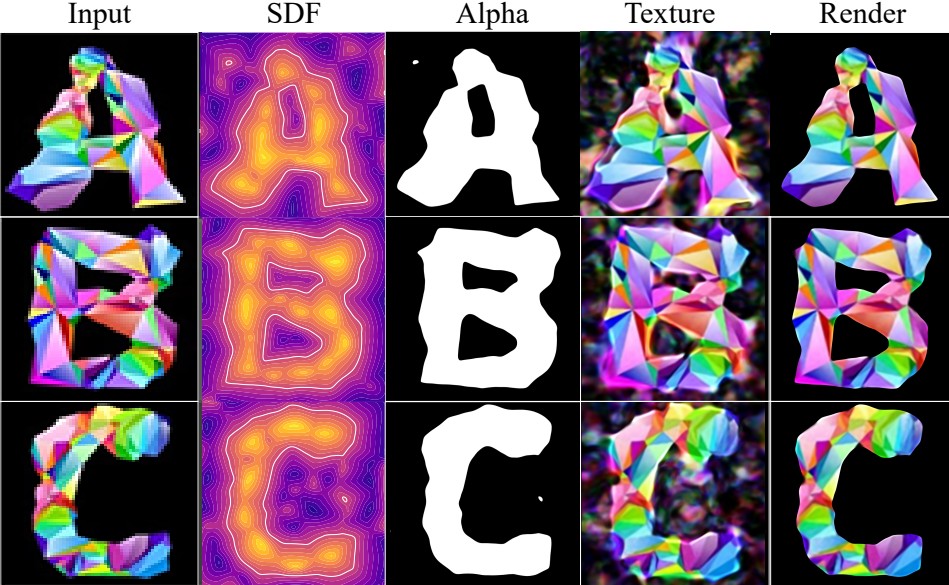

Figure S.6: **General Rasterization.** From left to right we present the input images, distance field estimate by an implicit neural network, alpha mask generated by function $K$ from Eq. S.1, font texture estimated by the implicit neural network $g$, final render created by multiplying the alpha mask and the texture.

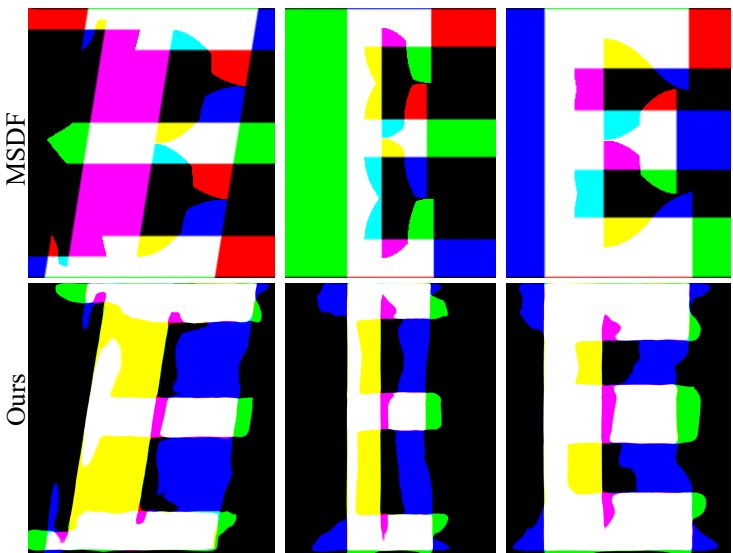

Figure S.7: **Canonical representation.** MSDF edge coloring based encoding does not have a canonical representation, therefore training a neural network with MSDF encoding based global supervision does not allow the network to learn a continuous latent space **(Top)**. Our local corner template supervision does not unnecessarily constraint the network thereby allowing our network to learn a consistent canonical multi-channel representation **(Bottom)**.

## S.9    More Quantitative Results

Learning a continuous interpolatable latent space is necessary to enable new font generation with our method. Therefore, after various experimentation, we selected LeakyReLU instead of Sine activation or Fourier features with our feed forward network. This can be quantitatively seen in the table below (similar to Table S.3 from the main paper). Here we calculate the interpolation metric like mentioned in Sec 4 of the main paper. We show the comparison between feed forward

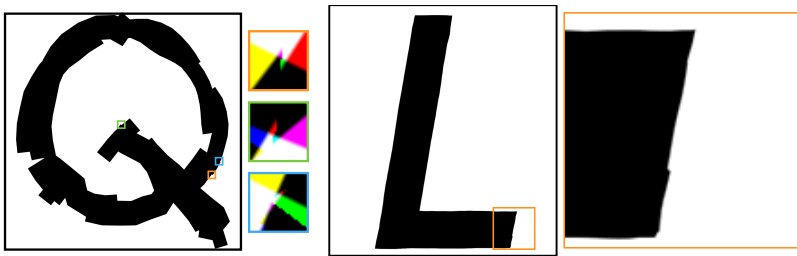

Figure S.8: **Limitations.** When a shape contains a small curve that ends in concave and convex corners the corner templates might overlap constraining the network to learn high-frequency details which are hard to capture by a neural network**(Left)**. Re-sampling the estimated multi-channel representation at an extremely high resolution will lead to wobbly lines and other undesirable artifacts that are not visible at the training resolution **(Right)**.

network with sine activation, Fourier features with bilinear resampling, Fourier features with implicit resampling and our method. All the methods are trained on a 50% subset of the dataset. Our method and Fourier features network are trained with a learning rate of $10^{-3}$ and sine activation network is trained with $10^{-5}$ learning rate. From the evaluations we conclude that Sine activation does not enable a continuous latent space. While Fourier features capture the shapes well their quality degrades at higher resolutions with bilinear resampling. Row 3 also shows that fourier features do not lend themselves well to implicit resampling.

Table S.3: Comparison with baselines on interpolations at different resolutions.

| Methods | MSE ↓ | | | | s-IoU ↑ | | | |
|---|---|---|---|---|---|---|---|---|
| | 128 | 256 | 512 | 1024 | 128 | 256 | 512 | 1024 |
| Sine Activation | .0857 | .08558 | .0864 | .0867 | .6191 | .6186 | .6199 | .6202 |
| FF Bilinear resampling | .0322 | .0344 | .0361 | .0368 | .8040 | .8041 | .8078 | .8089 |
| FF Implicit resampling | .2693 | .2747 | .3118 | .3413 | — | — | — | — |
| Ours | .0307 | .0315 | .0327 | .0331 | .8314 | .8318 | .8343 | .8350 |

## S.10  Training Warm-up

We have observed that it helps better convergence of font with relatively small features if initializing the aliasing range to the whole distance field range, i.e, (-1, 1), and slowly decreasing it to the desired values, using the mean function as $\hat{F}$. In addition, the learned distance field is relatively smoother. Figure S.9 visualizes the distance field estimated with and without the warm-up routine. When we compute the Laplacian to measure the smoothness of the estimated SDF, the SDF trained with warm-up has a value of 14.13, and the SDF estimated without warm-up has a value of 36.07. Note that there are many ways to achieve similar results, for example, explicitly adding the Laplacian to the loss terms will encourage the network to learn a smooth SDF estimate. Each of the alternatives has to choose between computationally cost and simplicity.