# OpenReview forum: "A Multi-Implicit Neural Representation for Fonts"
_NeurIPS.cc/2021/Conference — NeurIPS 2021 Poster_

### Official Review · Reviewer_iAuP · 2021-07-13

**Rating:** 4
**Confidence:** 4

**Summary:**

This paper proposes multi-implicit neural representation for fonts given the observation that it is hard to represent complex fonts with single neural representation. The author introduces a local supervision method that can train multi-implicit functions without knowing ground truth channels. Experimental results in various settings, e.g. reconstruction, interpolation, and generation, show that it outperformed the previous methods such as rasterized representation VAE and a couple of neural vectorized representations.

**Limitations And Societal Impact:**

Author addressed the limitations in the supplementary, my comments and suggestions are included in the main review.

I do not see any negative societal impact from this submission.

**Main Review:**

1. My main concern is that there is no clear experimental evidence that ‘multi-implicit’ functions are really necessary. Based on recent successful applications of neural implicit representations, e,g, NeRF and SIREN for 2d images, 3d shapes, etc, I do not see the reason why single implicit representation is not capable of generating complex fonts, and the author did not present the comparison against those. They mentioned the reason was SIREN and fourier feature papers do not learn a continuous latent space in the supplementary (L11-13). I do not think this is a valid reason, and in my opinion, it is very straightforward to just apply fourier features and sine activation function in the proposed method.

2. In my opinion, the proposed training techniques, such as local supervision, are specifically designed for this particular task (edge and corner aware sampling). Although it might not be a good reason to underestimate the work, but I think it is a limitation in terms of broader impact.

3. I encourage the authors to show ‘multi-implicit’ channels qualitative results in the main text. I could find a couple of examples in the supplementary, but it would also be great if the authors could explain more details about the results.

4. Multi-curve representation seems to be an interesting approach, but as I mentioned earlier, I do not think there are experimental results that support the effectiveness of the proposed method.

[minor comments]
1. L276-277: Did you train like GAN? random sampling ‘z’ during training? I think more explanation is required.
2. L213: edge/corner-award sampling -> edge/corner-aware sampling?
3. L186: what would be the format of glyph label ‘l’? it was mentioned in L276, but better to explain when it shows up for the first time.
4. How does interpolation work in experiment sections? linear interpolation between each ‘z’? then how did you get corresponding ‘z’ given the image.

[Update]
Post-Rebuttal: change the score 3 -> 4

**Time Spent Reviewing:**

4

---

> ### Author Response · Authors · 2021-08-10
> **Thank you!! response to concerns and baselines.**
>
> We would like to emphasize that the multi-implicit aims to better represent sharp corners (even after upscaling) when modeling thousands of instances at the same time, while single-implicit cannot reconstruct high fidelity/sharp corners if trained with the same samples as our multi-implicit. Existing works have demonstrated such drawbacks of single-implicit. Statistical results are shown above, and more visual comparison can be found in the links provided below.
>
> 1)  **Fourier features and Sine activation** have shown impressive reconstruction performance on fitting to a single instance. But to the best of our knowledge, Sine activation has been used to fit multiple instances only in pi-GAN and MetaSDF. However, the network that estimates latent variable z in pi-GAN is ReLU based and the network that estimates weights of the inner-network in MetaSDF is also ReLU based. We have extensively experimented with MetaSDF. We realized it is not possible to achieve high detail results on a large dataset in a zero-shot manner due to the smoothness constraint on the predictions of the outer network. Since this constraint is required for convergence, we choose to use a feed forward network instead.
> Earlier, we have also experimented with Fourier features, Sine activation and Sawtooth activation function with a feed forward network, due to the high frequency nature of the function the latent space learned by such a network is not continuous. Since learning a continuous latent space is essential for applications like font family generation from a complete or partial glyph, we resorted to using a LeakyReLU based activation function. Happy to include a discussion about the effect of choice of activation functions in the supplemental. Please refer to the rebuttal results link for more qualitative results. We have also added a more exhaustive evaluation to the general comments above. Which quantitatively shows the benefits of multi-implicit representations.
> 2) We believe C0 discontinuities are interesting, especially for fonts, and we propose a learning method to model C0 discontinuities using continuous functions with local supervision.
> 3).4.  We apologize for the confusion. We have shared an extensive collection of hand-picked and randomly sampled results with the supplementary. Due to the size of the results we have provided a link to the results in 20-21 of the supplementary. https://drive.google.com/file/d/1Dp-ECh7nulIkvMZw67fL3TBPyGsQHjM9/view?usp=sharing
> All the results contain the input, individual channel estimates, sdf and final render on reconstruction, interpolation and generation tasks. Hope this addresses your concern about evaluation and visual examples. Also refer to rebuttal results are available at https://drive.google.com/file/d/1Gzdq0eDC2uyM2OMF6HvkHdEYkmKis3if/view?usp=sharing
>
> Minor
> 1) We apologize for the confusion, as mentioned in Ln 2 of supplementary we train a auto-decoder network similar to DeepSDF we initialize \hat(z) to the mean of the z values learnt during training.
> 2). 3.    Thank you for the correction and we’ll update those in the final paper.
> 4. Yes we linearly interpolate the z values. We learn the z values during the training time.

---

> > ### Comment · Reviewer_iAuP · 2021-08-30
> > **Response to rebuttal**
> >
> > Hello, thanks for your response and also additional experiments.
> >
> > I agree with other reviewer's opinion that this work has multiple good contributions to the community, e.g. local supervision technique, etc. However, my concern is not fully resolved.
> >
> > The additional experimental results w/ fourier features were 'only for interpolation' tasks. For interpolation tasks, you compare 'a random interpolated output (generated by implicit neural nets)' to 'its nearest neighbour' in the training dataset. This can conclude that fourier features on ‘z’ does not enable a continuous latent space, which I can agree with. However, that was not my concern. My question was why ‘single implicit function’ w/ fourier feature can not generate ‘sharp corner’. In order to resolve my concern, you could have presented the ‘reconstruction task’ results.
> >
> > Also, we do not have to apply fourier features to 'z' if non-smoothness of latent space is the issue. We can only apply fourier feature to '(x,y)'.
> >
> > Based on all reviews and opinion, I would increase my score from 3->4. If this gets accepted, I would be very much appreciated for you to present additional experimental results in the final version, 1. on reconstruction task w/ fourier features and 2. fourier features to only '(x,y)' no on 'z'.

---

> > > ### Author Response · Authors · 2021-08-30
> > > **re: thank you and response**
> > >
> > > Thank you for the reply and helpful suggestions.
> > >
> > > 1. Sorry for the confusion, indeed Fourier features *recover* sharp corners in the reconstruction task. In the general comments, we *already acknowledged* that Fourier features encoding does capture a glyph well. The larger point we were trying to make is that since a feed-forward network with Fourier features encoding does not seem to learn a continuous latent space and hence it wouldn't enable applications like font family generation. Also, the quality of reconstruction degrades at higher resolutions with bilinear resampling and Fourier features encoding does not lend itself well to implicit resampling(shown in the rebuttal results). We agree to add these points and a reconstruction comparison to the final paper.
> > > 2. In the rebuttal experiments results, we apply Fourier features to the (x, y) coordinates *only* as suggested.
> > >
> > >
> > >
> > > To summarize, we believe that the experiments are in line with the reviewer's suggestion and we acknowledge the reviewer's point about reconstruction and justify why LeakyReLU activation is still an appropriate choice for the proposed architecture considering the use cases.
> > >
> > >
> > > We would also like to point out that the system does not seem to reflect the updated score.

---

> > > > ### Comment · Reviewer_iAuP · 2021-08-30
> > > > **re: re: thank you and response**
> > > >
> > > > Thanks for the quick response.
> > > >
> > > > 1. Could you provide quantitative results for the reconstruction task w/ fourier features? apology if this is too much request.
> > > >
> > > > 2. Out of curiosity, if you applied fourier features to only (x,y) coordinates, then why latent space 'z' is not learning the smooth latent space? is 'z' then same as your proposed model, right? any intuition on this?
> > > >
> > > > 3. I updated my score, sorry for the confusion.

---

> > > > > ### Author Response · Authors · 2021-08-31
> > > > > **response and qualitative results**
> > > > >
> > > > >
> > > > > 1. We report a quantitative evaluation of Fourier features versus ours for the corner reconstruction task.
> > > > > |              | MSE       |               |               |               |sIOU       |               |               |               |
> > > > > | ----------- | ----------- | ----------- | ----------- | ----------- | ----------- | ----------- | ----------- | ----------- |
> > > > > |Fourier Features Bilinear resampling|0.0045|0.0077|0.0101|0.0116|0.9013|0.9022|0.9021|0.9022|
> > > > > |**Ours**|0.0059|0.0082|0.0095|0.0101|0.9014|0.9020|0.9022|0.9022|
> > > > >
> > > > > 2. We believe Fourier features encoding makes it easier for the network to overfit. While overfitting is desired in some scenarios, it does not seem to interpolate or do resampling as desired. We also believe that a multi implicit representation allows the model to capture sharp changes in the final shape while letting the changes in individual channels be smooth. Intuitively this should give a model more flexibility to learn a smooth latent space between shapes. Also, we train using local supervision so as to not constrain the model unnecessarily.
> > > > >
> > > > > In https://drive.google.com/file/d/1xEQVGx-9ax7JveLutKDMr3GoKjpUH8zd/view?usp=sharing we share reconstructions of Fourier features encoding with bilateral resampling, implicit resampling, and ours multi-channel for qualitative comparison. The results show the Fourier features resampling artifacts as we mentioned. Hope that answers your questions. We would be happy to provide further details if requested.

---

> > > > > > ### Comment · Reviewer_iAuP · 2021-09-01
> > > > > > **thanks for your additional experiments!**
> > > > > >
> > > > > > Thanks for providing the reconstruction results. I really do appreciate it. I guess 64x64 training resolution might be the culprit that single implicit function w/ fourier features performed a bit lower. One obligatory experiments would be to add the fourier feature to 'multi implicit' to see if you can gain more.
> > > > > >
> > > > > > I appreciate the issue of smooth latent space, and I think it depends on the applications. For this particular application, I agree that fourier feature might discourage it, not sure why, and hoping that you could elaborate more in the final version of the paper if this submission gets accepted.
> > > > > >
> > > > > > I also encourage you to rewrite the main text in a way that the main issue is not about single implicit function's inability to represent 'sharp corners'. You could emphasize more on non-continuous latent space for generative models. For example,
> > > > > >
> > > > > > L40-43: 'Unfortunately, implicit representations (e.g., signed distance fields) for fonts are often too complex to be represented accurately by neural networks. As a results, although deep implicits work for simple fonts, they can fail to retain characteristic features (i.e., edges and corners) for complex fonts'
> > > > > > -> Considering fourier features and sine activation functions are very prevalent in 'implicit implicits', this might mislead the readers, which was my main concern.
> > > > > >
> > > > > > With all due respect, thanks for your invaluable works and I guess the more discussion would not be ideal. Again, I do appreciate your prompt response for my requests.

---

> > > > > > > ### Author Response · Authors · 2021-09-01
> > > > > > > **thank you and response**
> > > > > > >
> > > > > > > Thank you, we appreciate the comments and, given a chance, we will add the suggested discussion to the paper.

---

### Official Review · Reviewer_DnKK · 2021-07-15

**Rating:** 7
**Confidence:** 4

**Summary:**

This paper aims to learn a representation for rasterized fonts that allow for re-scaling, reconstruction, interpolation, and generation by preserving local features such as sharp corners or smooth curves. The proposed method composes multiple implicit representations (i.e., signed distance fields) to model complex details observed in neutral and stylistic fonts. The core idea is to represent corners as an intersection of multiple curves, enabling the construction of sharp corners in arbitrary resolutions. This assumption is further supported by a local supervision objective for the corners. In the experiments, the authors present that their model is able to preserve sharp corners. The proposed model outperforms baselines both qualitatively and quantitatively.

**Limitations And Societal Impact:**

The broader impact statement highlights the positive impacts. There may exist potential misuse of the proposed method. For example, learning and synthesizing a person’s handwriting for forgery. Would it be possible?

**Main Review:**

**Originality**
The idea of representing fonts as a multi-channel signed distance field (SDF) is proposed in [4]. This paper incorporates it into learned SDFs by using the recent techniques in neural implicit surfaces. While the core idea is not novel, this combination enables learning canonical representations across several font families and hence allows for interesting applications such as super-resolution, interpolation, and exemplar-based font generation.

**Quality**
The paper motivates the problem and the proposed method well. The paper focuses on learning a representation that can preserve the sharp corners in the downstream tasks. At the first glance, it does not sound like a challenging problem. The authors are able to illustrate the problem in the existing methods. The proposed approach for this problem is technically sound and its formulation seems correct. Experiments provide enough evidence showing the proposed method’s good performance compared to the selected baselines. The font completion results in Fig. 7 and the textured font results provided in the supplementary look impressive.

In my opinion, the main limitation of the submission is the baselines. The authors focus on font models and ignore other relevant baselines which leave some of the claims unsupported. Several SDF methods such as DeepSDF [1] and SIREN [2] could be used in the baselines to provide evidence for why multi-channel SDF is necessary. SDFs are very powerful and shown to be very effective in modeling complex 3D shapes. However, the authors claim otherwise in #41, “… implicit representations (e.g., signed distance fields) for fonts are often too complex to be represented accurately …”. I think the DeepSDF method could be a perfect baseline for this task. I am also wondering why the actual multi-channel SDF [4] is not considered in the reconstruction experiments. This would provide a piece of evidence for using a learning-based SDF.
In the ablations, it would be useful to see the contribution of design choices namely the corner loss, the sampling strategy, and using three channels instead of one (i.e., standard SDF with the corner loss and sampling).

Some claims are requiring further discussion or evidence.
- Related work Ln. 90-92: The proposed approach can also be categorized as a transformation from a raster to a continuous domain. How does it alleviate the problem of lack of finer details in the rasterized images?
- Related work Ln. 82-85: is there evidence for the inferior performance of sequential models compared to CNN-based models?
- Ln. 134 requires citation: “There are many works for estimating d(x; y) from scene parameters but most of them are constrained by the choice of the parameterization.”

**Clarity**
The overall organization of the paper is good. It is well-written. The illustration provided in Fig. 4 is very helpful for understanding the core idea.

**Significance**
It is an interesting problem, and the proposed method is very effective. It outperforms the baselines. However, in my opinion, the main problem is the selection of the baselines. It is not able to convince me if the proposed approach is really necessary. I am wondering whether a standard SDF could achieve similar results or not.

[1] Park et al., DeepSDF: Learning Continuous Signed Distance Functions for Shape Representation

[2] Sitzmann et al., Implicit Neural Representations with Periodic Activation Functions

-----------------------------------------------
**Updated:**
I thank the authors for their rebuttal and additional results supporting their submission. The clarifications and the new experiments show the advantage of the proposed multi-channel setup and the training objective in the raster domain. The local supervision seems to be useful in learning a canonical representation space, which is particularly helpful in modeling a large and diverse set of fonts. I raised my score by 1.

**Time Spent Reviewing:**

4

---

> ### Author Response · Authors · 2021-08-10
> **Thank you!! response to concerns and baselines.**
>
> Thank you for the suggestion about the additional baselines. We provide quantitative comparisons to the suggested baselines on a subset of our total data (see the common section). This bolsters our empirical argument that direct supervision like in DeepSDF (Fig 3 column 2) misses fine details, while Sine and Fourier features encoding do not enable latent space operations like font family generation in our case.
>
> Our novelty lies in training a **multi-implicit** network with only local corner supervision -- this is particularly useful for complex fonts (e.g., heavily stylized fonts like Fig 6). A method supervised using the multi-channel sdf approach proposed in [4] wouldn't learn desirable transformations between glyphs. This is because the output of [4] does **not** have a canonical form. We show an example in Fig S7. Again, this is an important advantage of local versus global supervision.
>
> Ln 90-92 We refer to C_0 discontinuities. In traditional vector graphics, rasterization pipelines we first calculate a min operation between SDFs of different curves in the shapes and rasterize the output. The min operation makes the process lossy. Once again, our local supervision helps us capture these C_0 discontinuities. But we understand the reviewer’s point that those are not the only lossy elements of rasterization, we will correct the text accordingly in the final draft.
>
> Ln 82-85 SVG-VAE and DeepSVG are the SOTA vector graphics sequential models. However we can see that DeepSVG struggles with font data from our comparisons, Im2vec comparisons and the DeepSVG results were presented on font data present in section F of their supplementary.
>
> Ln 134 we refer to pipelines like Im2Vec where the method is hardcoded to predict n number of bezier curve parameters, where n is a hyperparameter. In Fig 6b we show how such a constraint might be detrimental when we encode fonts with multiple curves and no straightforward correspondence between different curves of instances of the same character in the dataset.
>
> We have experimented with directly supervising with SDF values like in DeepSDF. For example the SDF regression error for the shape in Fig 3 column 2 is 0.0001 while the raster MSE is 0.0163 compared to column 4 SDF regression error 0.0884 and raster MSE 0.00002. Increasing the sampling rate might alleviate this problem but at the cost of computational time. We have noticed we can directly solve this problem by training in the raster domain. We have also added a more exhaustive evaluation to the general comments above. Which quantitatively shows the benefits of multi-implicit representations. For a qualitative comparison please refer to https://drive.google.com/file/d/1Gzdq0eDC2uyM2OMF6HvkHdEYkmKis3if/view?usp=sharing the benefits are especially apparent with characters like ‘E, F, M, N’ where there are many corners.
>
> We agree with the reviewer that if trained on a handwriting dataset such a method could possibly be used for emulating a person’s handwriting for forgery. We will add this observation to the social impact statement. Thank you.

---

> > ### Comment · Reviewer_DnKK · 2021-08-29
> > **Update**
> >
> > I thank the authors for their rebuttal and additional results supporting their submission. The clarifications and the new experiments show the advantage of the proposed multi-channel setup and the training objective in the raster domain. The local supervision seems to be useful in learning a canonical representation space, which is particularly helpful in modeling a large and diverse set of fonts. I raised my score by 1.

---

> > > ### Author Response · Authors · 2021-08-30
> > > **re: thank you and response**
> > >
> > > Thank you!! we will add the information from the discussion to the final version of the paper.

---

### Official Review · Reviewer_Ex1G · 2021-07-16

**Rating:** 7
**Confidence:** 4

**Summary:**

This paper deals with the problem of glyph generation in fontset design. Whereas previous works used raster-based or vector-based representations (and combinations of the two) to learn generative models for fonts, this paper proposes the use of what they call “multi-implicit representations”. In the proposed method, implicit functions (in the style of “NeRF”) are learned to represent subregions of a glyph by combining multiple simple shapes (eg two half-planes to create a corner). These subregion-implicit models are learned with local supervision, and a multi-Signed Distance Field model learns to compose them into the global glyph.


**Limitations And Societal Impact:**

Yes, although I discuss some missing limitations above

**Main Review:**

# Major Comments
The paper presents impressive results, including great handling of heavily stylized fonts (Fig 6).

The method requires differential rendering, as well as sampling from the individual regions. Can the authors comment on the computational complexity of training this model? Does the individual-glyph training complicate the computational expense further? Is sampling slow, since it has to sample the subregions and compose them? Does sampling scale well with resolution?

I am unsure on the utility of this method for designers. The output of the model is a combination of SDFs which need to be rendered in a custom renderer (within the model), and not a set of commands that can be manipulated in traditional font design software. This is in contrast with the papers that use vector graphics directly, which output a standard format. If this is true I believe it’s important to discuss the method’s limitations, including appropriate citations of methods that don’t have this limitation.

NeRF is trained for a single scene. Is that also true for each glyph/font? It seems like the paper uses a z latent space that was trained with the method in [17]. If that’s correct, it might be worth citing it on line 280.

# Minor Comments
How does it compare to “Learning implicit glyph shape representation?” I realize this is a new paper, but I’d be interested in reading a discussion on the differences of the two approaches
https://arxiv.org/abs/2106.08573v1

I am not as familiar with SDFs and implicit models as I am with font-generation models, so I defer the review of technical aspects of the implicit models to other reviewers

Overall, the method is really interesting and the results seem really good. If my major comments are resolved I would consider increasing the score by 1.


**Time Spent Reviewing:**

2

---

> ### Author Response · Authors · 2021-08-10
> **Thank you and response to concerns**
>
> We train all the glyphs and their associated latent vector z **concurrently** using mini-batch gradient descent (similar to other contemporary deep learning methods). The model took us close to 4 days to train on 2 TitanX GPUs. Since we cached important samples prior to training, we did not face any computational bottlenecks in terms of sampling at runtime. We note that importance samples scale quadratically w.r.t to the number of pixels. If we assume v to be the number of visible pixels and n the number of total pixels, importance samples scales O(v^2) compared to O(n^2) which is the case with non-implicit architectures.
>
> Yes, we can directly output vector graphics, if required. Given recent advances in vectorization, converting the SDF to vector graphics is straight-forward, especially since our multi-implicit representation allows probing at desired resolution. In Figures 5 and 6, we already showed vectorized output of our method -- vectorized in a piecewise linear manner as polylines. Please use digital zoom to see the differences in reconstruction and interpolation between ours and the baseline methods. One can simplify the piecewise linear outputs using quadratic fitting (using adaptive samples) or the simplify tool that is readily available in Illustrator or other popular vector graphics editing softwares. However, we did not explore this direction.
>
> ‘Learning Implicit glyphs’ is a concurrent work that was released after our paper submission. It is encouraging that a concurrent work also believes that font glyphs are better expressed as SDFs. However, they do not supervise using a raster loss which might leave the network susceptible to not capturing fine details like shown in Fig 3 column 2. This is also visible in their interpolation results. We propose to add a more in-depth discussion to the final paper.
>
> Thank you, we will add the suggested citations and rectify the spelling mistakes in the final version.

---

> > ### Comment · Reviewer_Ex1G · 2021-08-20
> > **re: rebuttal**
> >
> > Post-rebuttal comment: I thank the authors for the response to my comments. Thanks, also, for providing additional details about the computational complexity of sampling and how it scales quadratically, and about the process to vectorize the model’s output. I think these are important details that should be mentioned in the final version of the paper. Provided these are added, I consider my major comments resolved, and therefore raise my score by 1.

---

> > > ### Author Response · Authors · 2021-08-22
> > > **re: thank you and response**
> > >
> > > Thank you!! we will add the information from the discussion to the final version of the paper.

---

### Official Review · Reviewer_Fqtj · 2021-07-20

**Rating:** 7
**Confidence:** 3

**Summary:**

The paper proposes a method for representing vector graphic fonts with multi-implicit functions.
The main goal is to allow the reconstruction of fonts at any resolution without loss of discontinuities such as corners and edges.
Instead of representing a font shape as one SDF and then optimising the training process against this SDF, the paper proposes to represent a shape using multi-SDFs, and then differentiably rendering a shape against this multi-SDF representation and optimizing the input shape against this rendering. This allows the method to bypass the need for collecting a vast amount of vector graphics data for training.
The paper shows that this kind of multi-SDF representation helps in preserving discontinuities at corners and edges.

**Limitations And Societal Impact:**

Yes, the authors have made a comment about the positive societal aspect of the proposed approach.

**Main Review:**

Originality:
The problem being addressed in the paper is an important one. The proposed approach of multi-SDF representation of fonts is interesting and helpful for the problem (as evidenced by the experiments in the paper).

Quality:
The paper is a good read. It is well articulated and the goals are clearly stated. The paper compares exhaustively to prior works (both image-based and curve-based approaches) on different tasks such as high-resolution reconstruction, font style interpolation, font family completion, and glyph completion. Important ablation on the need for multi-SDF's to recover/preserve sharp corners when rendering fonts in high-resolution has been demonstrated using two resampling techniques -- implicit resampling (which densely resamples the implicit filed) and bilateral resampling (which upsamples the SDF prediction from the implicit model to the target resolution). The experiments show the effectiveness of the proposed approach in modeling vector graphic fonts, both in absolute terms and relatively.

Clarity:
Except for a few typos, the paper is well-written.
L36 - "resultant" to "resulting"
L62 - "evaluated" to "evaluate"
L97 - "as mentioned above" (instead of "as aforementioned")
L124 - same comment as above

Significance:
I think the problem being addressed is getting attention from the community for the last year. The paper follows this trend and makes a modest contribution to this line of research (which is okay).


**Time Spent Reviewing:**

3 hrs

---

> ### Author Response · Authors · 2021-08-10
> **Thank you!!**
>
> We appreciate the comments and positive encouragement, and agree to make the requested changes in the final version.

---

### Author Response · Authors · 2021-08-10
**Thanks for the reviews. Rebuttal response and baselines.**

We thank the reviewers for their precious feedback; given a chance, we will incorporate the suggestions into the final draft.
The main objective of our method is to perform generation of high detail font glyphs. We enable this by learning a latent space of font glyphs using global and local supervision in the raster domain.

Direct distance field supervision like in DeepSDF does **not** capture the fine details of shape well. This can be seen qualitatively in  Fig 3 Column 2 of the paper. Therefore, we chose to train a model in the raster domain instead of direct sdf supervision. Here we also present a quantitative evaluation of reconstruction loss in the corners (similar to Table S1 from supplemental) between deepsdf, our method single-channel and our method multi-channel. All the three methods are trained on a 50% subset of the dataset mentioned in the paper for 1000 epochs with a learning of 0.001. The table below quantitatively shows that raster supervision captures the shaper better than distance field supervision and multi-channel representation encodes the details better than the single channel representation. We also share an exhaustive sample of qualitative results in the link below. The effect of multi-channel implicit is more visible in characters like ‘E, F, M, N’ where there are many C0 discontinuities.


|               | MSE       |               |               |               |sIOU       |               |               |               |
| ----------- | ----------- | ----------- | ----------- | ----------- | ----------- | ----------- | ----------- | ----------- |
| DeepSDF | 0.0195|    0.0229|    0.0243 |    0.0250|0.8971    |      0.897|    0.8975|0.8975     |
|Ours Single-Channel|0.0072|0.0098|0.0111|0.0117|0.9012|0.9019|0.9020|0.9021|
|**Ours Multi-Channel**|**0.0059**|**0.0082**|**0.0095**|**0.0101**|**0.9015**|**0.9021**|**0.9022**|**0.9023**|

Learning a continuous interpolatable latent space is necessary to enable new font generation with our method. Therefore, after various experimentation, we selected LeakyReLU instead of Sine activation or Fourier features with our feed forward network. This can be quantitatively seen in the table below (similar to Table 2 from the main paper). Here we calculate the interpolation metric like mentioned in Sec 4 of the main paper. We show the comparison between feed forward network with sine activation, Fourier features with bilinear resampling, Fourier features with implicit resampling and our method. All the methods are trained on a 50% subset of the dataset. Our method and Fourier features network are trained with a learning rate of 10^-3 and sine activation network is trained with 10^-5 learning rate. From the evaluations we conclude that Sine activation does not enable a continuous latent space. While Fourier features capture the shapes well their quality degrades at higher resolutions with bilinear resampling. Row 3 also shows that fourier features do not lend themselves well to implicit resampling. We share more qualitative samples in the link below.

|               | MSE       |               |               |               |sIOU       |               |               |               |
| ----------- | ----------- | ----------- | ----------- | ----------- | ----------- | ----------- | ----------- | ----------- |
| Sine Activation| 0.0857|0.08558|0.0864|0.0867|0.6191|0.6186|0.6199|0.6202|
|Fourier Features Bilinear resampling|0.0322|0.0344|0.0361|0.0368|0.8040|0.8041|0.8078|0.8089|
|Fourier Features Implicit resampling|0.2693|0.2747|0.3118|0.3413|---|---|---|---|
|**Ours**|**0.0307**|**0.0315**|**0.0327**|**0.0331**|**0.8314**|**0.8318**|**0.8343**|**0.8350**|

We address individual concerns in the comments below. Rebuttal results: https://drive.google.com/file/d/1Gzdq0eDC2uyM2OMF6HvkHdEYkmKis3if/view?usp=sharing
All the glyphs in the link are sampled at (1024, 1024) resolution.

---

### Decision · Program_Chairs · 2021-09-27

**Decision:**

Accept (Poster)

**Comment:**

Meta-review of "A Multi-Implicit Neural Representation for Fonts"

The paper proposes representing vector fonts with multi-implicit functions. They were able to reconstruct vector fonts (at arbitrarily high resolution) while keeping specific font-discontinuities such as edges and corners, and also showcased their method to synthesize an entire font given one character only.

Reviewers were initially mixed. While reviewers generally agree that the proposed method works (and based on the results, seems to work really well), the concerns raised (esp by iAuP) is whether multi-implicit functions are indeed needed. DnKK raised a similar concern about whether standard SDF could attain similar results or not. The authors have responded to these concerns, and even produced some newer results in their response (They have also written a detailed general response with baseline comparisons). Reviewers were generally satisfied with their response, new results, and improvements made, and have increased their scores, and even persuaded the reviewer iAuP to improve their initial assessment.

In reviewer discussions, most reviewers are in favour of accepting the work. Reviewer iAuP strongly suggested that the authors clarify many issues they discussed in a detailed thread in the camera ready version. For instance, as they noted: *Given the author's additional experiments, it turns out that 'single implicts' w/ fourier features can reconstruct fonts as well as 'multi implicits', better in low resolution worse in high resolution, but roughly about the same.* So I would like to see the text clarified to the best of the authors' abilities (as they have pledged in the discussion thread) the text, and make sure the claims are accurate and clear, as to not mislead readers.

Given the generally high quality of the work, and fantastic results, which will be useful to the NeurIPS community and ML practitioners, I recommend accepting this work as a poster.